# TAPERED OFF-POLICY REINFORCE
# Stable and efficient reinforcement learning for LLMs

**Nicolas Le Roux**[*,▽,1]   **Marc G. Bellemare**[*,2]
Jonathan Lebensold[†,3]   Arnaud Bergeron[†,1]   Joshua Greaves[3]
Alex Fréchette[2]   Carolyne Pelletier[2]   Eric Thibodeau-Laufer[2]
Sándor Toth[2]   Sam Work[2]
[1]Mila   [2]Reliant AI

## Abstract

We propose a new algorithm for fine-tuning large language models using reinforcement learning. Tapered Off-Policy REINFORCE (TOPR) uses an asymmetric, tapered variant of importance sampling to speed up learning while maintaining stable learning dynamics, even without the use of KL regularization. TOPR can be applied in a fully offline fashion, allows the handling of positive and negative examples in a unified framework, and benefits from the implementational simplicity that is typical of Monte Carlo algorithms. We demonstrate the effectiveness of our approach with a series of experiments on the GSM8K and MATH reasoning benchmarks, finding performance gains for training both a model for solution generation as a generative verifier, and on a learning to search task, using the model as a query expander. We show that properly leveraging positive and negative examples alike in the off-policy regime simultaneously increases test-time accuracy and training data efficiency, all the while avoiding the "wasted inference" that comes with discarding negative examples. We find that this advantage persists over multiple iterations of training and can be amplified by dataset curation techniques, enabling us to match 70B-parameter model performance with 8B language models. As a corollary to this work, we find that REINFORCE's baseline parameter plays an important and unexpected role in defining dataset composition in the presence of negative examples, and is consequently critical in driving off-policy performance.

## 1 Introduction

Reinforcement learning (RL) and EM-type methods are rapidly becoming the dominant paradigm for fine-tuning LLMs on complex tasks such as chain-of-thought reasoning. These methods can amplify a base model's performance without additional human data and can optimize for synthetic rewards [53] and non-differentiable objectives [5]. While several popular methods rely solely on positive examples to fine-tune an LLM [52, 17], the "trial and error" nature of RL algorithms is especially well-positioned to leverage negative examples produced by the model, which are increasingly being recognized as key to efficient learning [39, 45, 54, 13]. In fact, there is mounting evidence that the simplest of all methods, REINFORCE [49], is a highly effective approach to fine-tuning LLMs [1].

However, REINFORCE is essentially an on-policy algorithm. In the presence of negative rewards, its good behaviour can only be guaranteed when the training data distribution (or reference model) matches, or is close to, the model's own distribution. This limits its ability to reuse past data, and

---

[*] Equal contribution.
[†] Equal contribution.
[▽] Corresponding author: nicolas@le-roux.name
[3] Work done at Reliant AI

39th Conference on Neural Information Processing Systems (NeurIPS 2025).

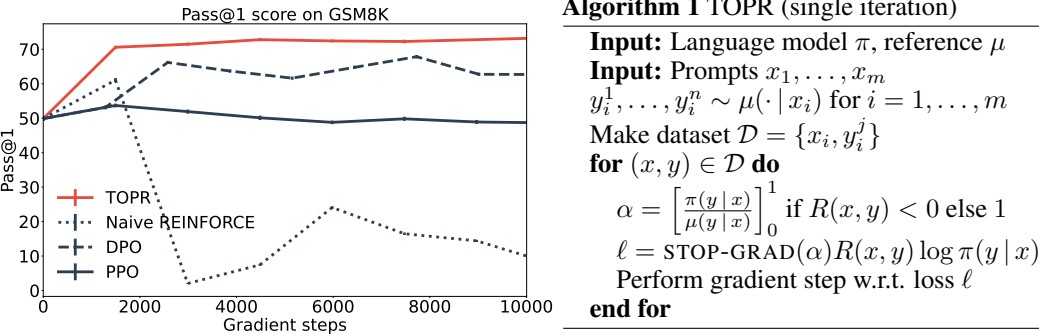

**Algorithm 1** TOPR (single iteration)

**Input:** Language model $\pi$, reference $\mu$
**Input:** Prompts $x_1, \ldots, x_m$
$y_i^1, \ldots, y_i^n \sim \mu(\cdot \,|\, x_i)$ for $i = 1, \ldots, m$
Make dataset $\mathcal{D} = \{x_i, y_i^j\}$
**for** $(x, y) \in \mathcal{D}$ **do**
$\quad \alpha = \left[ \frac{\pi(y \,|\, x)}{\mu(y \,|\, x)} \right]_0^1$ if $R(x, y) < 0$ else $1$
$\quad \ell = \text{STOP-GRAD}(\alpha) R(x, y) \log \pi(y \,|\, x)$
$\quad$ Perform gradient step w.r.t. loss $\ell$
**end for**

Figure 1: **Left: Test set accuracy (Pass@1) on the GSM8K benchmark [7] over the course of off-policy fine-tuning of the Llama 3 8B model.** As training becomes increasingly off-policy, the naive use of the REINFORCE gradient causes substantial performance degradation and PPO stops improving. DPO, which handles negative examples through a preference-based formulation, fares better but still falls well short of TOPR's performance. Pass@1 refers to the usual single-reasoning accuracy. See Section 4 for experimental details. **Right:** TOPR pseudo-code (Section 3).

puts pressure on the experimenter to select a just-right set of hyperparameters to avoid problems. Indeed, evidence of the instability of REINFORCE-type in off-policy training of LLMs can be found everywhere in the literature, from early work [36] to key algorithmic choices made in the training of the Kimi $\kappa 1.5$ [46] and DeepSeek-R1 [18] models. While KL regularization to the objective can mitigate this instability, it results in slower learning and requires additional hyperparameter tuning.

We propose *Tapered Off-Policy REINFORCE* (TOPR), a stable algorithm even when the model differs substantially from the data distribution, while fully leveraging positive and negative examples. TOPR improves a language model $\pi$ (or *policy*) by means of the asymmetric policy gradient

$$\nabla J_{\text{TOPR}}(\pi) = \sum_{\tau: R(\tau) > 0} \mu(\tau) R(\tau) \nabla \log \pi(\tau) + \sum_{\tau: R(\tau) < 0} \mu(\tau) \left[ \frac{\pi(\tau)}{\mu(\tau)} \right]_0^1 R(\tau) \nabla \log \pi(\tau) , \quad (1)$$

where $\tau$ is a response (or *trajectory*) sampled from some data-generating policy $\mu$, $R(\tau)$ is the reward associated with this trajectory, and $[x]_a^b = \max(\min(x, b), a)$ denotes the clipping function. The lack of importance ratio for positive examples means that TOPR will increase their probability, even when that importance ratio is small. The importance ratio is kept for negative examples as it ensures that the model will stop using capacity for them once their probability is small enough. Unlike other LLM algorithms in the REINFORCE family, TOPR does not require an explicit KL penalty to guarantee stable behaviour, making it both simpler to implement and computationally more efficient. Compared to PPO [40], DPO [39], and the "naive" application of REINFORCE [1], TOPR continues to improve reasoning performance even once $\pi$ differs substantially from $\mu$ (Figure 1, left).

We characterize the stable off-policy performance of TOPR by using it to train language models to reason within the GSM8K and MATH benchmarks. We use these benchmarks to highlight the various factors at play in off-policy reinforcement learning of language models, in particular the importance of positive/negative balance in the training dataset. Surprisingly, we find that REINFORCE's baseline parameter – commonly used as a variance reduction mechanism – plays an altogether different role of balancing the dataset in this regime, and is essential to good performance. Critically, our results indicate that choosing the best baseline requires taking more than only the mean return into account. We conclude with a series of multi-iteration experiments showing the ability of TOPR to fine-tune language models well beyond their base benchmark performance.

## 2 Off-policy policy optimization

We consider an autoregressive language model $\pi$ that, given a prompt $x$, assigns a probability to a length-$n$ response $y$ according to $\pi(y \,|\, x) = \prod_{i=1}^{n} \pi(y_i \,|\, x, y_{<i})$. Given a reward function $R(x, y)$

that measures the quality of the response $y$ to $x$ and a dataset of prompts $x_1, \ldots, x_m$, we wish to maximize the expected reward $J(\pi) = \frac{1}{m} \sum_{j=1}^{m} \left[ \mathbb{E}_{y \sim \pi(\cdot \mid x_j)} R(x_j, y) \right]$. In this paper, we abstract the prompt-response relationship and view this problem through the lens of policy optimization, where $\tau$ is a trajectory produced by the language model (i.e., the policy). With mild abuse of notation, we thus write $J(\pi) = \mathbb{E}_{\tau \sim \pi} R(\tau)$.

The original REINFORCE algorithm [49] maximizes $J(\pi)$ through the process of *on-policy policy optimization*. In the simplest form of the algorithm, a single trajectory $\tau$ is sampled according to $\pi$, and the parametrized policy $\pi$ is updated according to the unbiased gradient estimate $\nabla \hat{J}(\pi) = R(\tau) \nabla \log \pi(\tau)$, whose expectation is $\nabla J(\pi) = \mathbb{E}_{\tau \sim \pi} R(\tau) \nabla \log \pi(\tau)$.

In practice, training is rarely truly on-policy, for example because data is generated in a parallel, asynchronous fashion [33] or in a separate "sidecar" process [35, 18]. It is obviously also desirable to reuse trajectories throughout training, especially when generating these trajectories incurs a substantial computational cost or because they have been generated by a different process (e.g., expert trajectories). In the *off-policy policy optimization* setting, we assume the existence of a reference distribution $\mu$, typically different from $\pi$, which produces training trajectories. In the fully online case, which we mainly consider in this work, trajectories are sampled once and never updated. The goal is to find the optimal policy using this fixed set of trajectories. We also consider in our experiments the setting where trajectories are only rarely updated. Our main goal in this paper is to highlight the pitfalls of dealing with negatively-rewarded trajectories in off-policy policy optimization and propose a solution – TOPR – that avoids these pitfalls to produce performant, stable behaviour when training language models. By way of explaining the algorithmic choices behind TOPR, we review existing solutions and how they fall short of our desiderata.

## 2.1 The problem with naive REINFORCE

As a warm-up, consider a binary reward function $R(\tau) \in \{-1, 1\}$ and the algorithm that samples a trajectory $\tau$ from $\mu$, then updates the policy $\pi$ according to the REINFORCE update: $\nabla \hat{J}_{\mu}(\pi) = R(\tau) \nabla \log \pi(\tau)$. This essentially corresponds to the "naive" off-policy application of the REINFORCE update [1]. In expectation, this update maximizes the objective

$$J(\mu) = \sum_{\tau \in T^+} \mu(\tau) \log \pi(\tau) - \sum_{\tau \in T^-} \mu(\tau) \log \pi(\tau) \, , \tag{2}$$

where $T^+$ and $T^-$ are the set of trajectories with positive and negative rewards, respectively. The first term is maximized by making $\pi$ as close to $\mu$ as possible on the positive subset $T^+$. The second term, on the other hand, incentivizes $\pi$ being *as far from $\mu$ as possible*. This term is unbounded above (in terms of $\pi$) and can be made arbitrarily large by driving the probability of any single trajectory supported by $\mu$ to zero. This acts as a destructive force on the the model parameters, driving them to producing infinitely negative logits and without safeguards, eventually causes degenerate behaviour.[2]

We show in Section 3.1 that, while the issue can be mitigated by early stopping, the use of a baseline parameter, or KL regularization towards $\mu$, these modifications effectively work by fully or partially ignoring negative trajectories and thus limit the amount of learning that can be done off-policy.

## 2.2 Supervised fine-tuning

A simple solution to avoid the catastrophic failure of the model due to negative trajectories is to remove them from the dataset entirely. This can be interpreted as a form of reward-weighted supervised fine-tuning (SFT). The corresponding objective is

$$J_{\text{SFT}}(\pi) = \sum_{\tau \in T^+} \mu(\tau) R(\tau) \log \pi(\tau) \, ,$$

where the trajectory $\tau$ has weight $\mu(\tau) R(\tau)$ if $R(\tau)$ is positive, 0 otherwise. If we write $\mu_R^+(\tau) \propto \mu(\tau) R(\tau)$, then $-J_{\text{SFT}}(\pi)$ is the cross-entropy loss between $\mu_R^+$ and $\pi$.

---

[2]This issue doesn't appear in the on-policy application of REINFORCE because, by definition, a trajectory $\tau$ whose probability $\pi(\tau)$ is small is unlikely to arise in the dataset.

Supervised fine-tuning in the usual sense [56] can be viewed as the special case where all positive rewards are equal to +1 and $\mu$ is fixed and independent of the language model. STaR [52], ReST [17], and ReST-EM [43] use SFT with a dataset generated by the LLM itself, i.e. $\mu$ is equal or close to $\pi$, or by another LLM, possibly with a filtering step to further enhance dataset quality.

Removing negative examples from the dataset yields an objective that is bounded above, making these methods stable. As they are implemented with a cross-entropy loss, they can also quickly learn to mimic the distribution $\mu_R^+$, a characteristic that we retain in TOPR. However, omitting negative examples comes at a cost: for challenging problems, there may be few positive examples, and finding them may require additional machinery such as reference-guided grading [55], and wasted inference cycles. Mathematically, the lack of negative examples means that $\pi$ is incentivized to stay closer to $\mu$, limiting the amount of progress that can be achieved before having to resample from the LLM.

## 2.3 Truncated importance sampling

Importance sampling is perhaps the most common technique to address distribution shift. From $J(\pi) = \mathbb{E}_{\tau \sim \mu} \left[ \frac{\pi(\tau)}{\mu(\tau)} R(\tau) \right]$, we can derive an unbiased estimate of the on-policy gradient: $\nabla \hat{J}_{\mathrm{OPR}}(\pi) = \frac{\pi(\tau)}{\mu(\tau)} R(\tau) \nabla \log \pi(\tau)$. We call this the *off-policy REINFORCE* (OPR) gradient. In theory, that equation provides a convenient algorithm for optimizing the true objective $J(\pi)$: sample a trajectory $\tau \sim \mu$ and weight its update by the importance ratio $\frac{\pi(\tau)}{\mu(\tau)}$. In practice, it is well-known that importance sampling is plagued with excessive variance. This is problematic when optimizing over sequences, where the importance ratio is a product of many per-step ratios [38]. Gradient variance matters for positive trajectories – whose probability $\pi(\tau)$ increases during training – and negative trajectories, where a single excessive ratio can have a destructive effect on the model parameters.

One can mitigate the variance issue by truncating the importance ratios [34, 12, 14]. The corresponding sample gradient is $\nabla \hat{J}_{\mathrm{TIS}}(\pi) = \left[ \frac{\pi(\tau)}{\mu(\tau)} \right]_0^1 R(\tau) \nabla \log \pi(\tau)$.

Truncated importance sampling (TIS) is an integral part of TOPR. There are, however, situations where following the gradient of $J(\pi)$ is *not desirable*, justifying further enhancements. To see this, note that when the importance ratio $\frac{\pi(\tau)}{\mu(\tau)}$ is close to 0, so is the norm of the gradient. Should this happen for trajectories with positive rewards, the model will take a long time to increase that trajectory's probability. This is not specific to importance sampling and is an issue with the usual on-policy REINFORCE [22]. We shall see in our experiments how TIS, while effective, is more sensitive to dataset composition and the choice of reward baseline.

## 2.4 PPO and other methods

PPO [40], one of the most widely used policy-based methods, optimizes the objective $J_{\mathrm{PPO}}(\pi) = \mathbb{E}_{\tau \sim \mu} \min \left( \frac{\pi(\tau)}{\mu(\tau)} R(\tau), \left[ \frac{\pi(\tau)}{\mu(\tau)} \right]_{1-\epsilon}^{1+\epsilon} R(\tau) \right)$ for $\epsilon \in (0, 1)$. This objective implements an asymmetric treatment of positive and negative rewards and is essentially composed of three parts (Fig. 2). In the near on-policy setting, when only few updates are made before resampling trajectories, this can be quite effective; GRPO [18], for example, modifies $J_{\mathrm{PPO}}$ with a batch-dependent baseline.

However, the PPO objective applies the importance ratio to the reward rather than to the gradient. As a consequence, the gradient of $J_{\mathrm{PPO}}(\pi)$ becomes zero outside of the $[1 - \epsilon, 1 + \epsilon]$ range, limiting its usefulness and potentially causing brittleness when more than a handful of updates are made before resampling trajectories. The algorithm is also not incentivized to reduce the relative probability of negative trajectories below $1 - \epsilon$, limiting the potential improvement from $\mu$. Although variants such as sPPO increase this robustness [47], their performance still drops after many updates.

We focused here on methods that work with batch size 1, i.e. that do not need to compare examples. We discuss other methods, including popular ones like DPO, in Appendix A.

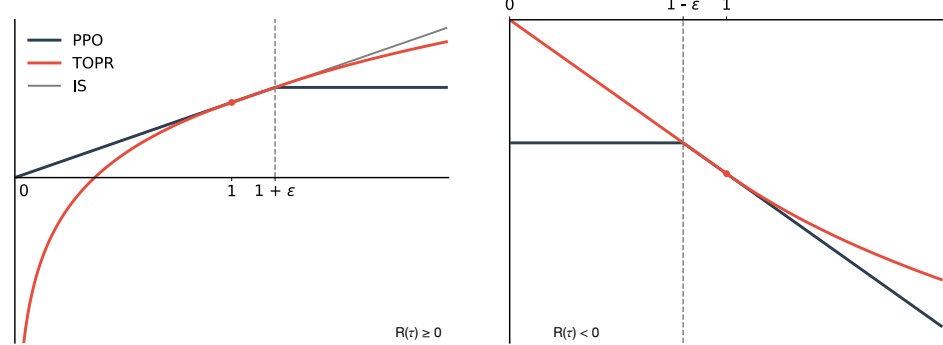

Figure 2: **Visualization the TOPR objective (Eq. 6 with $a^+ = 1, b^+ = 1 + \epsilon$, $a^- = 0, b^- = 1$) contrasted with PPO, as a function of the importance ratio $\frac{\pi}{\mu}$ (IS).** The two losses are equal on the intervals $[1, 1 + \epsilon]$ and $[1 - \epsilon, 1]$ (positive and negative rewards, respectively). However, PPO stops gradients when the ratio differs substantially from 1, which prevents it from being an effective off-policy algorithm. TOPR also implements a sharper positive-example loss for small importance ratios, accelerating the learning of these examples.

| | $a^+$ | $b^+$ | $a^-$ | $b^-$ | Negative examples | Bounded objective | Low variance | Fast learning |
|---:|:---:|:---:|:---:|:---:|:---:|:---:|:---:|:---:|
| SFT | 1 | 1 | 0 | 0 | No | **Yes** | **Yes** | **Yes** |
| Naive REINFORCE | 1 | 1 | 1 | 1 | **Yes** | No | **Yes** | **Yes** |
| Off-policy REINFORCE | 0 | $+\infty$ | 0 | $+\infty$ | **Yes** | **Yes** | No | No |
| Truncated IS | 0 | 1 | 0 | 1 | **Yes** | **Yes** | **Yes** | No |
| TOPR | 1 | 1 | 0 | 1 | **Yes** | **Yes** | **Yes** | **Yes** |

Table 1: TOPR combines the advantages of supervised fine-tuning, REINFORCE, and importance sampling to support stable and efficient off-policy fine-tuning of language models.

## 3 TOPR: Tapered off-policy REINFORCE

We now introduce the TOPR algorithm. TOPR uses importance sampling to downweight negative trajectories that are not likely under $\pi$, while allowing positive trajectories to be upweighted irrespective of $\pi$. The framework we consider involves two sets of truncation limits, $a^+ \leq b^+$ and $a^- \leq b^-$:

$$\nabla J(\pi) = \sum_{\tau \in T^+} \mu(\tau) \left[\frac{\pi(\tau)}{\mu(\tau)}\right]_{a^+}^{b^+} R(\tau) \nabla \log \pi(\tau) + \sum_{\tau \in T^-} \mu(\tau) \left[\frac{\pi(\tau)}{\mu(\tau)}\right]_{a^-}^{b^-} R(\tau) \nabla \log \pi(\tau) . \quad (3)$$

By choosing different truncation limits, we obtain many of the methods introduced in the previous section (Table 1). TOPR itself corresponds to a range of truncation limits that combine the desirable properties of each of these methods into one learning rule.

**Gracefully unlearning negative trajectories.** Setting $a^- = 0$ allows the algorithm to progressively reduce the contribution of negative trajectories, as provided by importance sampling. Any $a^- > 0$ must eventually lead to model degeneracy as with naive REINFORCE (Section 2.1).

**Quickly learning positive trajectories.** Setting $a^+ > 0$ gives the benefits of supervised fine-tuning: we ensure a minimum rate of learning for positive trajectories and accelerate their learning when they have a low probability under $\pi$. This allows us to avoid the "quasi-local minima" issue that plagues REINFORCE in high-dimensional action spaces.

**Trading off bias and variance.** The upper truncation limits allow us to keep gradient variance under control, as expected from truncated importance sampling. This is important early in training for negative examples, when a few examples may exhibit a very large importance ratio, and late in training for positive examples, where we expect the untruncated ratio to be much greater than 1.

We follow Occam's principle in defining the canonical form of TOPR as the algorithm where $a^- = 0$ and all other parameters are 1. This yields the expected TOPR gradient:

$$\nabla J_{\text{TOPR}}(\pi) = \underbrace{\sum_{\tau \in T^+} \mu(\tau) R(\tau) \nabla \log \pi(\tau)}_{\text{SFT update for positive examples}} + \underbrace{\sum_{\tau \in T^-} \mu(\tau) \left[ \frac{\pi(\tau)}{\mu(\tau)} \right]_0^1 R(\tau) \nabla \log \pi(\tau)}_{\text{TIS update for negative examples}}, \quad (4)$$

which combines the SFT update for positive examples, leading to acceleration, and the TIS update for negative examples, allowing for their handling without brittleness. Algorithm 1 sketches out an implementation of TOPR in an off-policy, deep learning setting.

We will demonstrate in Section 4 that this canonical parametrization is highly effective and provides robustness to the choice of data distribution and deep learning hyperparameters. Before doing so, however, we provide theoretical justification for the design choices behind TOPR.

## 3.1 Analysis

In Section 2.1 we argued that introducing a baseline parameter cannot create stable off-policy learning behaviour without risking sacrificing performance. We will make this point more precisely in this section and the next. To begin, let us revisit the expected naive REINFORCE update, now introducing a baseline parameter $c \in \mathbb{R}$:

$$\nabla J_{\mu,c}(\pi) = \mathbb{E}_{\tau \sim \mu} \left[ \left( R(\tau) - c \right) \nabla \log \pi(\tau) \right]. \quad (5)$$

The following establishes the contribution of positive and negative examples as well as the baseline to the expected loss $J_{\mu,c}(\pi)$.

**Proposition 3.1.** $\mathcal{L}_{\mu,c}(\pi)$, defined as $\mathcal{L}_{\mu,c}(\pi) = -J_{\mu,c}(\pi)$, is the four-part loss

$$\mathcal{L}_{\mu,c}(\pi) = C + R_\mu^+ KL(\mu_R^+ \| \pi) - R_\mu^- KL(\mu_R^- \| \pi) - c KL(\mu \| \pi),$$

where $\mu_R^-$ is the reward-weighted distribution

$$\mu_R^-(\tau) = \begin{cases} \frac{\mu(\tau)|R(\tau)|}{R_\mu^-} & \text{if } R(\tau) < 0, \\ 0 & \text{otherwise;} \end{cases} \qquad R_\mu^- = \sum_{\tau \in T^-} \mu(\tau) |R(\tau)|; \qquad T^- = \{\tau : R(\tau) < 0\},$$

and symmetrically for $\mu_R^+$, and $C$ is a constant independent of $\pi$.

Proposition 3.1 shows that the baseline induces KL regularization towards ($c < 0$) or away ($c > 0$) from the sampling distribution [see also 25, 47]. At $\mu = \pi$, we recover that the baseline has no effect on the expected on-policy gradient [49, 44]. In particular, when all rewards are positive ($R_\mu^- = 0$), Eq. 5 moves the policy $\pi$ towards a reward-weighted version of the sampling distribution $\mu$ [15].

Proposition 3.1 also shows that adding a baseline to minimize the impact of negative rewards ($c < 0$) works by regularizing $\pi$ towards $\mu$. To guarantee stable behaviour, the baseline must in general match the smallest negative reward (e.g., $R'(\tau) = R(\tau) - c \geq 0$). At this point, however, the baseline effectively removes negative trajectories from the objective function – losing the information contained in these trajectories. Using importance sampling instead avoids this issue. To see this, we start by noting that Eq. 3 is the gradient of

$$J_{\text{TOPR}}(\pi) = \sum_{\tau \in T^+} \mu(\tau) \rho \left( \frac{\pi(\tau)}{\mu(\tau)}, a^+, b^+ \right) R(\tau) + \sum_{\tau \in T^-} \mu(\tau) \rho \left( \frac{\pi(\tau)}{\mu(\tau)}, a^-, b^- \right) R(\tau), \quad (6)$$

where $\rho(\cdot, a, b) : [0, \infty) \to \mathbb{R}$ is the *taper function*

$$\rho(x, a, b) = \begin{cases} a \left( 1 + \log \frac{x}{a} \right) & \text{if } x < a \\ b \left( 1 + \log \frac{x}{b} \right) & \text{if } x > b \\ x & \text{otherwise.} \end{cases}$$

The taper function $\rho$ describes the effect of the truncation on the objective optimized by TOPR. It defines a lower bound on the importance ratio, in the sense that for any $a \leq b$,

$$\rho\left(\frac{\pi(\tau)}{\mu(\tau)}, a, b\right) \leq \frac{\pi(\tau)}{\mu(\tau)} ,$$

and it is equal to this ratio on the interval $[a, b]$ (Fig. 2). For our choice of $a^+ = b^+ = 1$ and a positive reward function $R(\tau) \geq 0$, TOPR optimizes a lower bound on the true objective $J(\pi)$:

$$J(\pi) \geq J_{\text{TOPR}}(\pi) = \mathop{\mathbb{E}}_{\tau \sim \mu}\left[\rho\left(\frac{\pi(\tau)}{\mu(\tau)}, 1, 1\right) R(\tau)\right] .$$

The bound follows from the analysis of related algorithms by [8, 25, 26] and [17]. The following proposition establishes the stable off-policy behaviour of TOPR for a wider range of truncation parameters.

**Proposition 3.2.** *For $a^- = 0$, Eq. 6 is bounded above: there exists a $B$ such that*

$$\sup_\pi J_{\text{TOPR}}(\pi) \leq B.$$

*Furthermore, for any $a^- > 0$, $J_{\text{TOPR}}(\pi)$ is unbounded above unless $R(\tau) \geq 0$ for all $\tau$.*

For positive examples, the taper function with $a^+ > 0$ maintains a substantial gradient even when $\pi(\tau)$ is small, since the weight used for the gradient of the surrogate objective, $\mu(\tau)$, is independent of the current policy $\pi$. This allows the model to recover from a low probability $\pi(\tau)$ assigned to a good trajectory, avoiding a traditional failure of REINFORCE. Empirically, [26] observed lower variance and more efficient learning when optimizing this log-ratio lower bound. With negative rewards, however, replacing the importance ratio with the log-ratio leads to the surrogate objective being an *upper bound* on $J(\pi)$ [25], which is a different way of expressing the conclusion of Proposition 3.2.

## 4   Results

We study the effectiveness of TOPR at training language models to perform chain-of-thought (CoT) reasoning, as well as preliminary results on fine-tuning an LLM to perform query expansion with the end goal of improving search results. For the most part we focus on the single-iteration, fully offline regime aiming to characterize the relative stability and effectiveness of TOPR for training language models compared to prior alternatives. Our results are naturally complementary to the full gamut of methods that improve language models iteratively.

Our main results are on mathematical reasoning datasets that require step-by-step solutions and are widely used to evaluate the reasoning capabilities of LLMs: GSM8K [7] and MATH [20]. As our core model, we use the Llama 3 family of instruction-tuned language models [10], using the 8B model unless otherwise specified. More details about the datasets and experimental setup may be found in Appendix C.

### 4.1   Fine-tuning chain-of-thought reasoning

Our first set of experiments aims to answer the question: *Is a more careful handling of importance ratios and rewards beneficial in off-policy policy optimization?* We begin with a comparison of PPO, DPO, naive REINFORCE, and TOPR. For DPO, pairs of candidate solutions are formed from these so as to obtain up to 16 contrastive pairs. For PPO, we use $\epsilon = 0.2$.

Fig. 1 (left) experimentally demonstrates the limitations of existing methods. Naive REINFORCE performs well at first but, in the absence of a KL term, collapses as $\pi$ moves away from $\mu$. As expected, PPO makes little progress on the objective as most data points quickly fall outside of its $[1 - \epsilon, 1 + \epsilon]$ range. DPO performs well off-policy, confirming the observations of [35], especially when measured in terms of Pass@1 accuracy. TOPR outperforms all these methods. We explore in more detail in Appendix D.1 some reasons for TOPR's success.

**TOPR minimizes reasoning failures.**   To understand the reasons behind TOPR's success, we measured the proportion of generated solutions that were correct, incorrect, or invalid (the string "The answer is" is not present) during the course of training. Fig. 7 gives strong evidence as to the

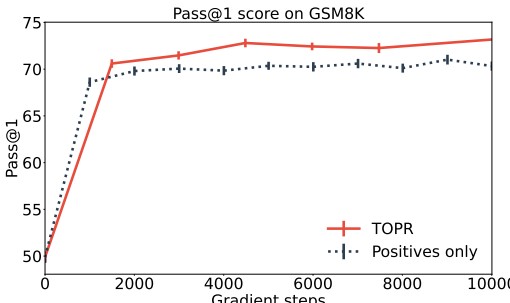

Figure 3: Test set accuracy on GSM8K across training when using all examples (TOPR) or positive examples alone. Not shown here, TOPR also yields higher inference efficiency at test time: greater Maj@N performance for all N.

root cause of REINFORCE's poor performance, whose generations are overwhelmingly degenerate by the end of training. By contrast, TOPR proves effective at teaching the model to avoid incorrect formatting – yielding the desirable property that one can solely rely on RL for solution generation, rather than using additional tools to correctly format them.

**Using negative examples improve performance.** To understand the impact of negative examples on training, we next formed a "positives only" dataset by removing all negative examples from our base dataset. This procedure mimics some of the design choices of recent work such as STaR that apply SFT as the inner loop of an RL-like procedure. Using this dataset results in stable learning but substantially lower performance than that of TOPR (Fig. 3). This translates into greater self-consistency [48] efficiency: more solutions must be generated at test time to reach the same level of performance (top right). A more detailed analysis of this experiment can be found in Appendix D.2.

**Striking the right balance of positive and negative examples.** We now refine the previous analysis by varying the effective proportion of positive examples in the dataset. We start with a dataset of 50,000 examples, 10% of which are positive. We then vary the baseline for each model to reach an effective proportion of positive examples from 1% all the way to 100%. Details on how changing the baseline can affect the effective proportion of positive examples may be found in Appendix B.

Fig. 4 shows the test performance on GSM8K using either TOPR or TIS when the effective proportion of positive examples varies. The optimal effective proportion to be around 10-20% for both datasets (GSM8K and MATH) and both algorithms (TIS and TOPR). We posit that a good baseline is one that achieves such a proportion. We also performed additional experiments in Appendix D.3 showing that this *effective* proportion is independent of the *actual* proportion of examples in the training set.

Our result also gives further evidence that the optimal baseline is not always the expected return in practical settings, contrary to common belief.[3]

**Acceleration improves robustness to dataset composition.** Given the important performance gains from incorporating negative examples to training and the relevance of dataset composition (Section B), it is natural to ask whether TOPR's positive-example acceleration ($a^+ = 1$) helps when positive examples are outnumbered in the dataset, for example because the problem at hand is very hard. When that proportion is low, the model tends to lower the probability of most trajectories in its training set, leading to the probability of positive trajectories being lowered too. Fig. 4 shows that, thanks to its acceleration ($a^+ = 1$), TOPR recovers from these cases while TIS cannot. When the effective proportion of positive examples is high, there is virtually no difference between TOPR and TIS. We also see that TIS reaches a slightly higher maximum Pass@1 accuracy (chosen over all experiments) compared to TOPR. This suggests that TIS may trade robustness for peak performance.

**Ratio truncation improves stability.** A natural question is whether the truncation of importance ratios as done by TOPR is necessary, or if standard importance sampling alone suffices, as suggested by [1]. To study this question, we trained from the base model as before but using standard importance sampling ($a^+ = a^- = 0, b^+ = b^- = +\infty$). Results and analysis may be found in Appendix D.4.

**TOPR outperforms across multiple iterations.** As a final experiment, we combine insights from our previous experiments to demonstrate that TOPR is an effective inner-loop algorithm for iterated

---

[3][37] and [8] remark on similar findings in the on-policy setting; see also the empirical study by [6].

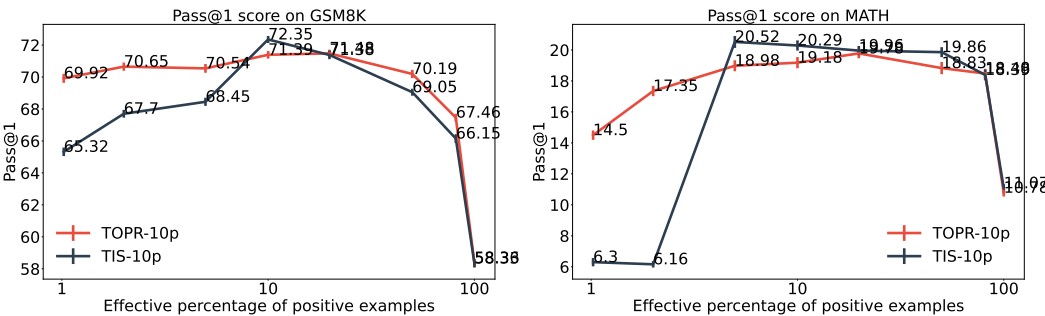

Figure 4: Test set accuracy on GSM8K (left) and MATH (right) when the training set contains $p = 10\%$ of positive examples, using either truncated importance sampling (TIS) or TOPR. The x-axis, on a log scale, represents the effective proportion of positives.

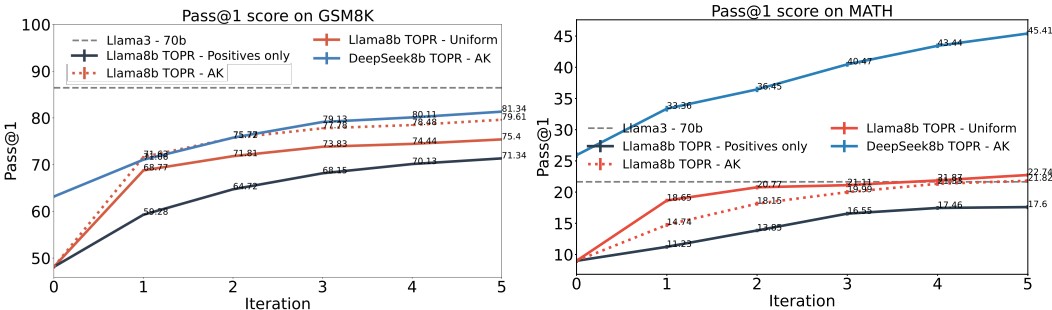

Figure 5: Pass@1 scores on GSM8K (left) and MATH (right) for uniform sampling, positive-only sampling, and Anna Karenina sampling (see main text). By combining TOPR and Anna Karenina sampling, a fine-tuned DeepSeek-R1 8B model outperforms Llama 3 70B.

fine-tuning of language models. Training begins with a base model $\pi_0$, from which a dataset is sampled ($\mu_i = \pi_{i-1}, i = 1, 2, \dots$). We then subsample this dataset to create an iteration batch of to $N = 50,000$ data points, and use these to train a new policy $\pi_i$ starting from $\pi_{i-1}$ as a reference. Figure 5 shows how model performance continues to improve over multiple iterations, both for GSM8K and MATH; furthermore, TOPR enables faster learning than positive-only sampling, allowing us to surpass DeepSeek 8B-level Maj@16 accuracy within a few iterations.

One challenge with fine-tuning models that already perform quite well is that, as training progresses, the model is essentially presented with examples that it already performs quite well on. This can make training quite inefficient. To combat this, we introduce a dataset-balancing technique we call *Anna Karenina sampling*, based on Tolstoy's famous "All happy families are alike; each unhappy family is unhappy in its own way." For each problem, we sample 64 candidate solutions, of which we only keep the *first* positive example. The iteration batch is then filled with negative examples chosen at random from those candidates. On GSM8K, this technique enables more efficient learning (**79.6%** Pass@1 accuracy) compared to uniform sampling (**75.4%**). It is less effective on early MATH iterations, where the model has low Pass@1 accuracy and every positive example counts. As further evidence of TOPR's effectiveness, applied to the more recent DeepSeek 8B model [18] it produces a model that rapidly surpasses the 70B version of Llama 3 in Maj@16 accuracy.

## 4.2 Learning to search

We now turn to preliminary results of TOPR's performance on a learning to search task. We apply reinforcement learning to replicate responses from a black-box API. Many systems that support text-based search rely on synonym and abbreviation expansion, where a user's query is automatically improved using a large corpus of morphological variants for the same concept [21]. While model-based approaches to expanding a user's query have been proposed [27, 9, 51], none of them have used RL to learn a better covering for user queries.

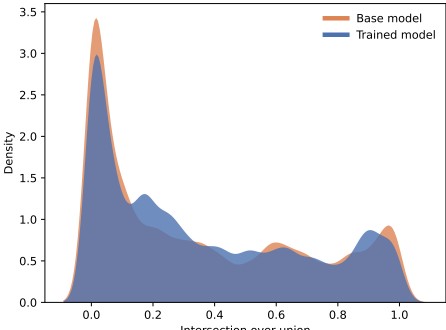

Figure 6: Test performance as intersection over union with TOPR (trained model). The trained model offers more set coverage where the base model fails to produce useful query expansions. Values below 0.0 are due to the smoothing and should be ignored.

For this purpose we consider the `clinicaltrials.gov` repository of clinical trial protocols (CTG). We studied whether a large language model can learn to produce a set of synonyms (an expansion) in response to a given query that minimizes the symmetric set distance between the CTG API and a simple internal database containing the same documents but differently indexed.

We used the Llama3-70B model to generate 1,200 distinct in-domain queries related to the field of life sciences, then defined a train-test split of 1,000 and 200 respectively. For each query, we prompted the smaller 8B model to propose 8 variations of up to 10 query expansions resulting in a training dataset of 8,000 samples to fine-tune our generator. Expansions were assigned a reward of 1 if they resulted in the same search results, or up to 5% relative tolerance for searches with 11 or more results.

Figure 6 shows the distribution of search result overlap for the test set, before and after training. We measure this overlap in terms of the intersection over union (IOU) metric, which as the name implies measures the ratio of shared articles to total articles. Thus, a score of 1.0 indicates a perfect match, and 0.0, completely different results. We find that, despite the challenging nature of this task, TOPR is able to improve on the base model by reducing the number of complete misses (score: 0.0). The average test IOU was improved from 35.03% (base model) to 35.83% (trained model). While preliminary, this result points to the potential value of using reinforcement learning to optimize language models to act as agents performing complex, but narrow tasks.

## 5   Conclusion and future work

Our results show that a simple but principled change to REINFORCE is all that is needed to deploy it successfully and stably in the off-policy regime. Our approach is more efficient than existing dataset curation methods: when generating the dataset as all data points are kept; at training time because no KL regularization is required, and negative examples are effectively made use of to improve performance; and at test time, because fewer solutions need to be generated. Our theory further provides an alternative, optimization-based perspective on truncated importance sampling for RL, which may warrant revisiting other algorithms that make use of it [34]. Finally, our analysis sheds new light on the role of the baseline parameter and dataset composition in off-policy RL.

From here, there are a number of future avenues for research. On one hand, we limited our experimental work to the setting where $\mu$ is the model at the beginning of an iteration, and data points are generated in the "self-taught" style. It would be beneficial to deploy this in the offline setting [28] with a different $\mu$, but at first glance this poses numerical challenges. We also limited ourselves to the training of large language models but there is no reason to believe that TOPR would not perform equally in other application areas of RL, from video games [3] to robotics [23].

## Author contributions and acknowledgments

NLR and MGB developed the algorithm, provided the theoretical analysis, structured the experimental work, and led the overall project. NLR, MGB, JL, AB, and JG implemented the algorithm, the experimental framework, and performed experiments. NLR, MGB, and JL wrote the paper. JL, JG, AF, CP, ETL, ST, and SW contributed to the technical infrastructure (datasets and software) with which the experimental work was performed.

NLR did this work while working as a consultant for Reliant AI, using their compute infrastructure. This work was partially supported by a Canada CIFAR AI Chair.

The authors would like to thank Matthieu Geist, Sal Candido, Rishabh Agarwal, Michael Bowling, Jesse Farebrother, Nathan Rahn, Harley Wiltzer, Charline Le Lan, and John Schulman for feedback and helpful discussions, as well as Matt Leus, Karl Moritz Hermann, and Nasib Naimi for support on this project.

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

## A  DPO and and other contrastive losses

We focused in the main text on methods that could work with a batch size of 1. This excludes all contrastive losses, which usually use pairs of examples, one positive and one negative. DPO [39], one of the most well-known of these methods, works with pairs of trajectories and maximizes the weighted log probability ratio of these two trajectories. For positive and negative trajectories $\tau_w$ and $\tau_l$, respectively, the DPO objective is

$$J_{\text{DPO}}(\pi) = \log \sigma \left( \beta \log \frac{\pi(\tau_w)}{\mu(\tau_w)} - \beta \log \frac{\pi(\tau_l)}{\mu(\tau_l)} \right),$$

where $\sigma$ is the sigmoid function. When rewards are either -1 or 1, DPO can be repurposed to handle negative and positive trajectories [19, 4], and is in fact well-suited to off-policy policy optimization [35]. However, DPO does not directly aim to maximize $J(\pi)$ and, with a finite number of trajectories, it is possible for the objective to increase while the probability of the positive trajectory decreases, as long as the probability of the negative trajectory decreases more. We shall see in our experiments that, while DPO indeed performs well off-policy, it is largely outperformed by TOPR. More recently, CoPG [13] also applied the idea of contrasting negatives and positives and, although their update shares a similar form with REINFORCE, the use of a carefully crafted baseline makes the method similar to DPO and IPO [2] and we therefore omit it from our analysis.

## B  Changing the effective positive rate through the baseline

In addition to the choice of the loss, the composition of the training set is critical to the performance of the trained model. In the context of training language models to perform chain-of-thought reasoning, for example, dataset curation methods such as STaR, ReST, and ReST-EM differ mainly on which data they include.

As we will see, the relative importance of positive and negative examples in the dataset is equally critical to good performance. Interestingly enough, the baseline parameter can also be interpreted as modulating this relative importance.

Let us again assume a binary reward function $R_0(\tau) \in \{-1, 1\}$ and a baseline $c \in [-1, 1]$. Let $p = |T^+|/(|T^+| + |T^-|)$ be the proportion of positive examples in the dataset. Substituting $R(\tau) = R_0(\tau) - c$ into Eq. 6, we obtain

$$J_{\text{TOPR}}(\pi) = \sum_{\tau \in T^+} \mu(\tau)\rho \left( \frac{\pi(\tau)}{\mu(\tau)}, a^+, b^+ \right) (1 - c) + \sum_{\tau \in T^-} \mu(\tau)\rho \left( \frac{\pi(\tau)}{\mu(\tau)}, a^-, b^- \right) (-1 - c)$$

$$= (1 - c) \sum_{\tau \in T^+} \mu(\tau)\rho \left( \frac{\pi(\tau)}{\mu(\tau)}, a^+, b^+ \right) - (1 + c) \sum_{\tau \in T^-} \mu(\tau)\rho \left( \frac{\pi(\tau)}{\mu(\tau)}, a^-, b^- \right).$$

With this transformation, the contribution of each positive example to the objective is weighted by $1 - c$. With some algebra, we find that the *effective* proportion of positive examples changes from $p$ to

$$\tilde{p} = \frac{p(1 - c)}{p(1 - c) + (1 - p)(1 + c)} = \frac{p(1 - c)}{1 + (1 - 2p)c}. \tag{7}$$

With a fixed dataset, we can thus vary the relative importance of positive and negative examples ($\tilde{p}$ and $1 - \tilde{p}$) by modifying the baseline according to Eq. 7. This generalizes the result from the previous section that discarding negative examples is equivalent to using a baseline of $-1$.

Furthermore, the choice of the baseline $c$ can be viewed as adding a softer version of the $\text{KL}(\mu \,\|\, \pi)$ term to the TOPR objective, again encouraging $\pi$ to stay close to $\mu$ when $c < 0$ (see Appendix F). As a negative baseline $c$ also increases the effective proportion of positive examples in the dataset, we see that a larger proportion of positive examples will decrease the degree of off-policyness that is achievable without resampling the training set. Adding negative examples can thus be seen as a way to further improve the policy.

It is however important to highlight some differences between setting a baseline and weighting positive and negative examples differently. While the expected gradient is the same, stochastic estimates can be different, which might impact techniques like gradient clipping or second-order optimization. We expect this difference to be larger for extreme values of the baselines.

We shall show in Section 4 how carefully choosing the effective proportion of positive examples, either through dataset composition or a baseline, can lead to a boost in accuracy.

## C  Datasets and experimental setup

### C.1  Models and Datasets

We focus on mathematical reasoning datasets that require step-by-step solutions and are widely used to evaluate the reasoning capabilities of LLMs. As our core model, we use the Llama 3 family of instruction-tuned language models [10], using the 8B model unless otherwise specified.

**GSM8K [7]**    The GSM8K dataset is composed of short grade-school math problems, requiring basic arithmetic or elementary algebra to solve. It contains 1,319 problems for testing and 7,473 for training. Verifying the correctness of model responses is straightforward, as the final answer is typically an integer. When the string is not present, we consider the answer as missing. For each training question we generate $n = 16$ candidate solutions using chain-of-thought (CoT) prompting, using the 8-shot prompt from [48]. We parse our model's answer by looking for the magic string "The answer is", matching this few-shot prompt.

**MATH [20]**    The MATH dataset contains problems from high school math competitions, covering a wide range of topics such as algebra, geometry, and probability, and is generally harder than GSM8K. We use the split from the original work, which includes 7,500 training problems. For computational reasons, we report performance on the smaller MATH-500 test set [30]. Each problem includes a step-by-step solution, ending in a final answer marked by $\boxed{}$ in the solution (e.g., "*..so the smallest possible value of c is $\boxed{\pi}$*"). This marking allows for verification of the correctness of model-generated responses by comparing the final answer to the ground truth. We use the script provided by [41] for this purpose.

For each training question we generate $n = 32$ candidate solutions using chain-of-thought (CoT) prompting, using the 4-shot prompt from [29].

### C.2  Experimental setup

Our training infrastructure is based on HuggingFace's transformers library [50]. We use data parallelism on a single H100 node with a per-GPU batch size of 1, a constant learning rate of 5e-7 chosen from a small parameter sweep, the Adafactor optimizer [42] to minimize memory usage, and neither weight decay nor KL regularization.[4] We divide the loss by the sequence length [16], which in reinforcement learning terms can be thought of as implementing hyperbolic discounting. We use HuggingFace's default gradient clipping parameter of 1.0. The reward is +1 for a correct answer and -1 for an incorrect answer, and no baseline is used. Candidate generations are produced using vLLM [24] with temperature $T = 1$, $\text{top}_p = 1$, $\text{top}_k = 500$, and a maximum of 512 tokens.

An iteration of training consists of generating candidates using a model (usually the base model), labelling those candidates with their associated reward, and performing a single epoch over this generated dataset. The reference $\mu$ corresponds to the model predictions at the beginning of the iteration. Unless otherwise noted, all reported scores and accuracies are with respect to test sets, measured by evaluating the model at the end of the iteration. No early stopping is performed. We use the bootstrap technique [11] to provide confidence intervals: we generate 64 solutions for each question for GSM8K and 16 for MATH due to the cost of evaluating. For each question, we select K answers at random without replacement, then compute the average maj@K or pass@K performance across the dataset. We repeat this process 100 times and estimated the empirical variance $\hat{V}$ across

---

[4]In addition to allowing us to focus on the relative stability of different learning rules, the removal of the KL term decreases the memory and computational burden of training the language model.

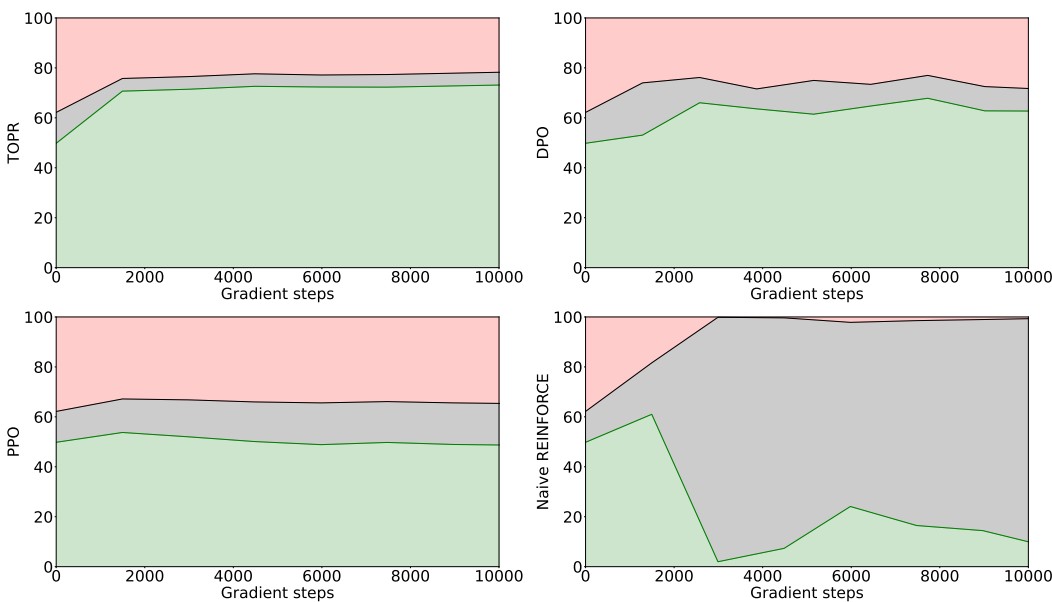

Figure 7: **Proportion of correct (green), incorrect (gray), and invalid (red) generated solutions on the GSM8K test set.** Of the four, TOPR is the only one to significantly reduce invalid generations.

the 100 trials. We compute the standard error as $\sqrt{\frac{V}{\frac{64}{K}-1}} = \sqrt{\frac{KV}{64-K}}$; depicted confidence intervals correspond to two standard errors.

## D    Additional experimental results

### D.1    TOPR minimizes reasoning failures

To understand the reasons behind TOPR's success, we measured the proportion of generated solutions that were correct, incorrect, or invalid (the string "The answer is" is not present) during the course of training. Fig. 7 gives strong evidence as to the root cause of REINFORCE's poor performance, whose generations are overwhelmingly degenerate by the end of training. By contrast, TOPR proves effective at teaching the model to avoid incorrect formatting – yielding the desirable property that one can solely rely on RL for solution generation, rather than using additional tools to correctly format them.

### D.2    TOPR vs. Positives only

Fig. 8 allows us to further analyze the experiment comparing TOPR, which uses both positives and negatives, to SFT which only uses the positive examples. Breaking down the test results as a function of the number of rationales that conclude in the correct answer ("Correct answer cardinality", bottom left), we find that TOPR's performance gains from using negative examples can be attributed to reducing the number of questions for which no or few solutions are found, guaranteeing a strong majority for self-consistency. We find similar results on MATH, where using TOPR enables us to almost double the Pass@1 accuracy compared to the base model (bottom right). Beyond these results, it is also worthwhile noting that TOPR is more *training inference-efficient*: indeed, because all vLLM generations are used to improve the model, training data is effectively generated as a faster rate.

### D.3    Effective vs actual proportion of positive examples

We observed in Sec. 4.1 that the optimal effective positive rate for GSM8K and MATH was in the 10%-20% range. These results, however, were obtained with a dataset that contained 10% of actual positive examples. One might thus wonder if the optimal performance is obtained when the effective

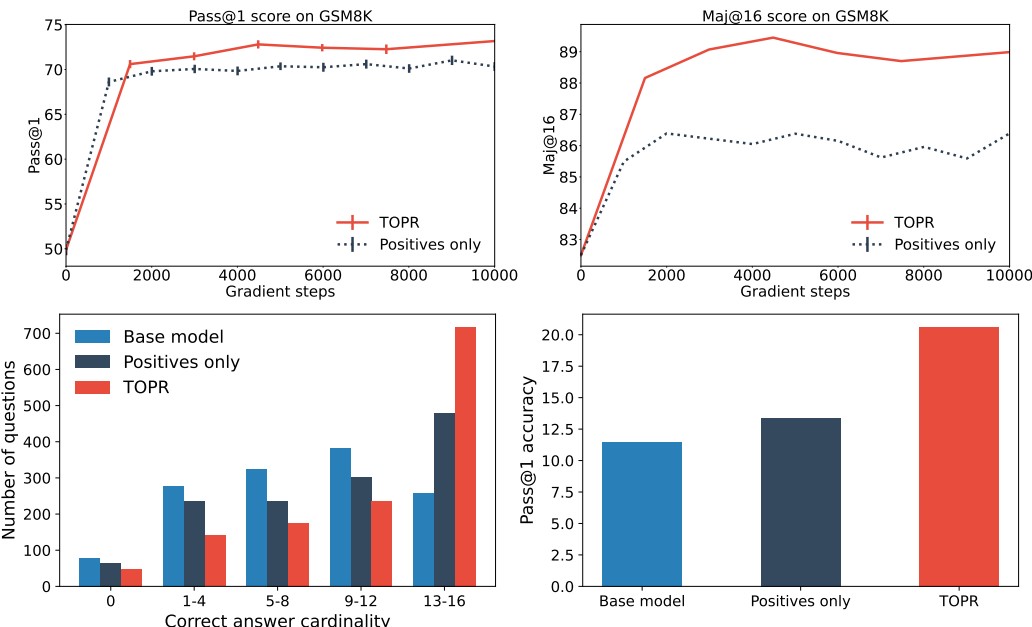

Figure 8: **Top** Test set accuracy on GSM8K across training when using all examples (TOPR) or positive examples alone. Not shown here, TOPR also yields higher inference efficiency at test time: greater maj@n performance for all n. **Bottom left:** Distribution of GSM8K problem questions as a function of the number of correct generations. TOPR more effectively reduces the number of questions with none (0) to few (1–4) correct generations. **Bottom right:** Pass@1 accuracy on MATH.

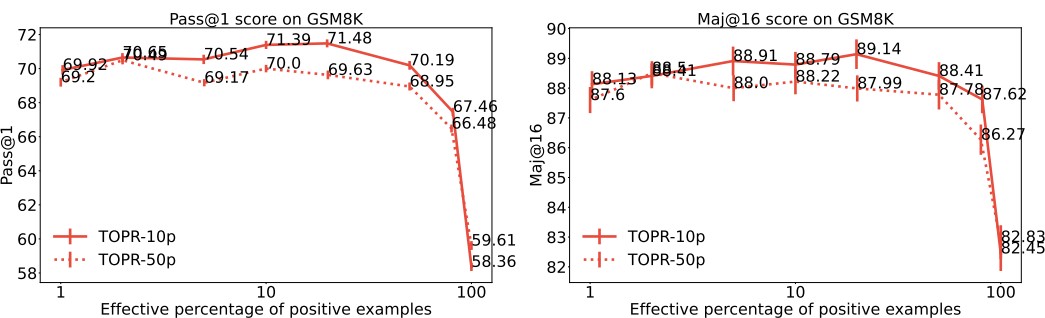

Figure 9: Test set accuracy on the GSM8K dataset when the training set contains either $p = 10\%$ (solid line) or $p = 50\%$ (dotted line) of positive examples as the baseline $c$ is varied. The x-axis, on a log scale, represents the effective proportion of positives $\tilde{p} = \frac{p(1-c)}{1+c(1-2p)}$. Pass@1 (left) and Maj@16 (right) results are shown.

positive rate matches that actual positive rate, i.e. with a baseline $c = 0$. For that purpose, in addition to the dataset containing 10% actual positive samples (labelled "10p"), we built one containing 50% of positive examples ("50p"). We then vary the baseline for each model to reach an effective proportion of positive examples from 1% all the way to 100%. We see in Figure 9 that, once again, TOPR's performance is maximal around 10-20% of effective positive examples, regardless of the *actual* proportion of positive samples in the training. Further, we observe a strong correlation between the 10% and 50% curves, showing that the effective proportion is a more critical factor than the actual proportion of positive samples. Further, Fig. 4 shows the optimal effective proportion to be around 10-20% for both GSM8K and MATH. We posit that a good baseline is one that achieves such a proportion.

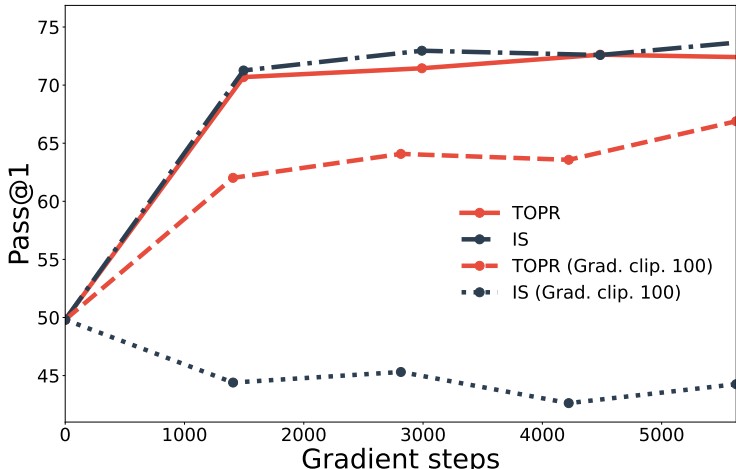

Figure 10: **Performance of TOPR and standard importance sampling (IS)** in our default experimental setting and with a higher gradient clipping parameter (100.0). TOPR shows greater robustness to the gradient clipping parameter.

**Acceleration improves robustness to dataset composition.** Given the important performance gains from incorporating negative examples to training and the relevance of dataset composition (Section B), it is natural to ask whether TOPR's positive-example acceleration ($a^+ = 1$) helps when positive examples are outnumbered in the dataset, for example because the problem at hand is very hard. Figure 4 shows the test performance on GSM8K using either TOPR or TIS when the effective proportion of positive examples varies. When that proportion is low, the model tends to lower the probability of most trajectories in its training set. This leads to the probability of positive trajectories being lowered as well. Thanks to its acceleration ($a^+ = 1$), TOPR recovers from these cases while TIS cannot. When the effective proportion of positive examples is high, there is virtually no difference between TOPR and TIS. Interestingly, we see that TIS reaches a slightly higher maximum Pass@1 accuracy (chosen over all experiments) compared to TOPR. This suggests that TIS may trade robustness for peak performance.

### D.4 The importance of truncating the importance ratio

Fig. 10 shows that this results in an algorithm that is as performant as TOPR. When we inspected its in-training behaviour, however, we found that the average gradient norm became increasingly larger as training became more and more off-policy. The impact of this norm blow-up is mitigated by the use of the default gradient clipping parameter (1.0), as well as the relatively low number of negative examples in the training dataset ($\sim$33%). To further demonstrate the stabilizing effect of ratio truncation in TOPR, we used the same two algorithms but now with a negatively-skewed dataset (60% negatives) and the gradient clipping parameter set to 100.0. The "Grad. Clip. 100" curves depict these results. TOPR is affected by these changes but still improves on the base model. Standard importance sampling, on the other hand, harms the model's performance – producing **31%** of bad reasonings by the end of training against **12%** for the base model.

### D.5 Experiments on Learning to verify

[53] and [32] recently studied the use of multiple CoT generations to verify the output of an LLM. In the context of math reasoning benchmarks, this *generative verifier* acts as a reward model that is used in a best-of-n selection scheme, improving performance over self-consistency [48]. Our next series of experiments aimed to study whether *TOPR can improve verifier performance and improve solution quality for harder problems*.

For each training sample in the MATH dataset, we used the 70B model to generate 16 solutions per problem, each with 4 verifications. We then fine-tuned an 8B model using TOPR to act as a generative verifier [53] using a total of 480,000 data points. We evaluated this generative verifier

| Verifier | Verifier accuracy | Invalid rate | Weighted SC |
|---|---|---|---|
| **None** | – | – | 55.5% |
| **Llama 3 8B** | 32.6% | 34.2% | 56.7% |
| **8B TOPR** | **70.9%** | **0.90%** | **61.5%** |

Table 2: **Performance of a verifier trained with TOPR compared to using a base model verifier or no verifier.** Here "Verifier accuracy" is the Pass@1 accuracy of the verifier (whether it correctly judges that a solution is right), "Invalid rate" is the number of verifications that could not be parsed successfully, and "Weighted SC" is weighted self-consistency with 32 generations (see e.g. [31]). When no verifier is used, Weighted SC indicates the usual maj@32 accuracy.

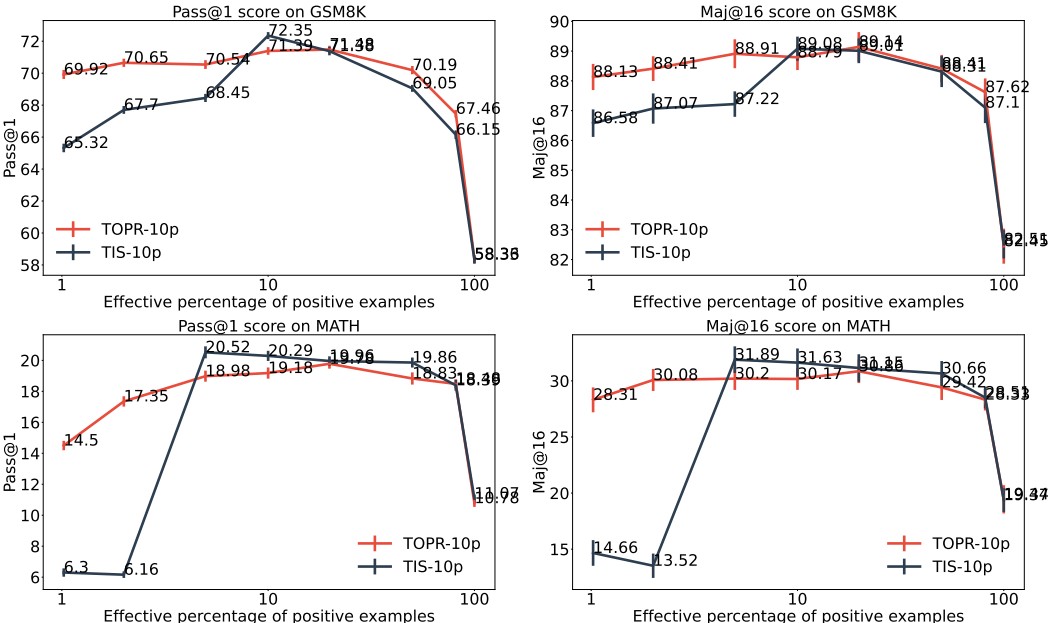

Figure 11: **Test set accuracy on GSM8K (top) and MATH (bottom) when the training set contains $p = 10\%$ of positive examples, using either truncated importance sampling (TIS) or TOPR.** The x-axis, on a log scale, represents the effective proportion of positives $\tilde{p} = \frac{p(1-c)}{1+c(1-2p)}$.

in a weighted self-consistency setting, where four verifications are aggregated into a score for each solution, and the answer with the highest score sum is selected. Table 2 shows that this procedure indeed produces a much more effective verifier for MATH generations, both in terms of its verifier accuracy and effect on solution quality. In an even more pronounced version of the results from Fig. 7, we find that TOPR fine-tunes the model to output almost no invalid generations – simply because it is negatively rewarded for doing so.

# E Additional Maj@16 results

We computed Maj@16 results in addition to the Pass@1 scores from the main text. Similar conclusions can be reached, whether it is about the comparison between TOPR and TIS when varying the effective positive rate (Fig. 11) or the performance of TOPR when performing multiple iterations (Fig. 12).

# F The impact of the baseline on KL regularization in TOPR

In standard REINFORCE, adding a baseline to the reward does not change the unbiasedness of the gradient estimate but affects its variance. The same is not true in off-policy policy optimization.

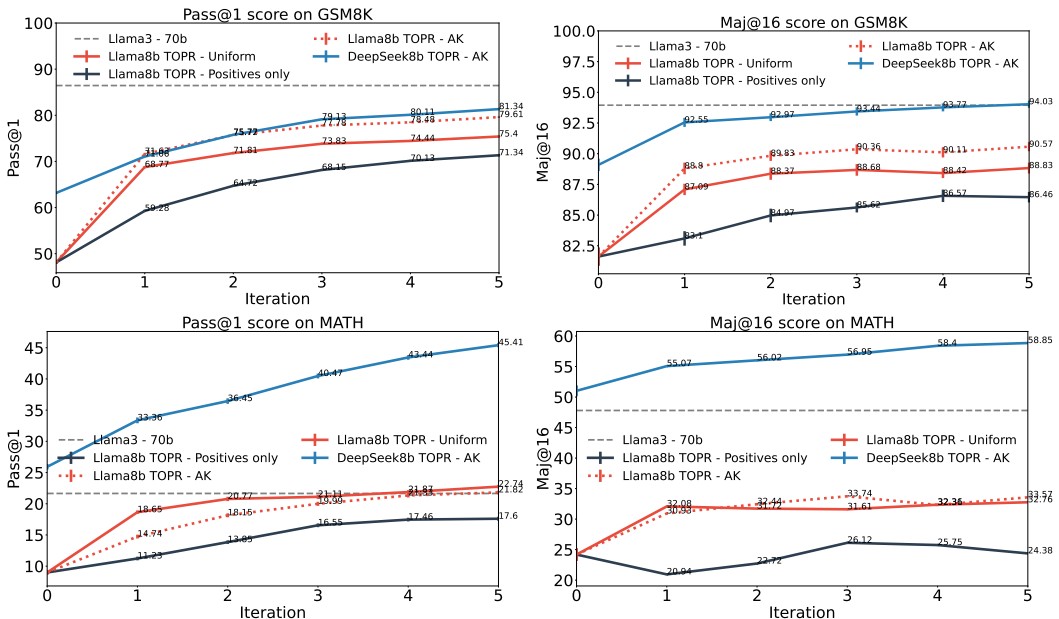

Figure 12: **Pass@1 (left) and Maj@16 (right) scores on GSM8K (top) and MATH (bottom) for uniform sampling, positive-only sampling, and Anna Karenina sampling (see main text).** By combining TOPR and Anna Karenina sampling, we are able to fine-tune the DeepSeek-R1 8B model to achieve performance slightly superior to Llama 3 70B.

Looking back at Eq. (3), assuming $a^+ = a^- = 0$, $b^+ = b^- = b$, we see that adding a baseline $c$ yields

$$
\nabla J_{\text{TOPR}}(\pi, c) = \sum_\tau \mu(\tau) \left[ \frac{\pi(\tau)}{\mu(\tau)} \right]_0^b [R(\tau) - c] \nabla \log \pi(\tau)
$$

$$
= \sum_\tau \mu(\tau) \left[ \frac{\pi(\tau)}{\mu(\tau)} \right]_0^b R(\tau) \nabla \log \pi(\tau)
$$

$$
- \sum_\tau \mu(\tau) \left[ \frac{\pi(\tau)}{\mu(\tau)} \right]_0^b c \nabla \log \pi(\tau)
$$

$$
= \nabla J_{\text{TOPR}}(\pi, 0)
$$

$$
- c \sum_\tau \mu(\tau) \frac{\pi(\tau)}{\mu(\tau)} \nabla \log \pi(\tau)
$$

$$
- c \sum_\tau \mu(\tau) \left( \left[ \frac{\pi(\tau)}{\mu(\tau)} \right]_0^b - \frac{\pi(\tau)}{\mu(\tau)} \right) \nabla \log \pi(\tau)
$$

$$
\nabla J_{\text{TOPR}}(\pi, c) = \nabla J_{\text{TOPR}}(\pi, 0) + c \sum_\tau \mu(\tau) \left( \frac{\pi(\tau)}{\mu(\tau)} - \left[ \frac{\pi(\tau)}{\mu(\tau)} \right]_0^b \right) \nabla \log \pi(\tau)
$$

Assume a negative baseline $c$. Since $\frac{\pi(\tau)}{\mu(\tau)} - \left[ \frac{\pi(\tau)}{\mu(\tau)} \right]_0^1$ is positive when $\pi(\tau) > b\mu(\tau)$, the additional term will *decrease* $\pi(\tau)$ in that case. Hence, a negative baseline will discourage $\pi(\tau)$ to be above $b\mu(\tau)$ for all trajectories $\tau$. In that sense, it acts as a softer version of a KL regularizer that only alters $\pi$ when it deviates too much from $\mu$. Alternatively, a positive baseline will encourage $\pi$ to be large, making the policy more deterministic. Note that this effect is solely due to clipping and goes against the effect due to stochasticity [6].

## G   Theoretical proofs

### G.1   Proof of Proposition 3.1

As in the main text, we separate positive and negative trajectories as $T^+ := \{\tau : R(\tau) \geq 0\}$ and $T^- := \{\tau : R(\tau) < 0\}$. Define the reward-weighted distribution

$$\mu_R^-(\tau) = \begin{cases} \frac{\mu(\tau)|R(\tau)|}{R_\mu^-} & \text{if } R(\tau) < 0, \\ 0 & \text{otherwise;} \end{cases} \qquad R_\mu^- = \sum_{\tau \in T^-} \mu(\tau)|R(\tau)|,$$

and symmetrically for $\mu_R^+$ and $R_\mu^+$.[5] We have

$$J_{\mu,c}(\pi) - J(\mu) = \mathop{\mathbb{E}}_{\tau \sim \mu} \left[ R(\tau) - c \right] \log \frac{\pi(\tau)}{\mu(\tau)}$$

$$= \mathop{\mathbb{E}}_{\tau \sim \mu} \left[ R(\tau) \log \frac{\pi(\tau)}{\mu(\tau)} \right] - c \mathop{\mathbb{E}}_{\tau \sim \mu} \left[ \log \frac{\pi(\tau)}{\mu(\tau)} \right]$$

$$= \sum_\tau \mu(\tau) R(\tau) \log \pi(\tau) + c \mathrm{KL}(\mu \,\|\, \pi) + C_0,$$

where $\mathrm{KL}(\mu \,\|\, \pi)$ denotes the Kullback-Leibler divergence from $\mu$ to $\pi$ and $C_0 \in \mathbb{R}$. We now break the first term of the above equation into its positive and negative components:

$$\sum_\tau \mu(\tau) R(\tau) \log \pi(\tau) = \sum_{\tau \in T^+} \mu(\tau) R(\tau) \log \pi(\tau) + \sum_{\tau \in T^-} \mu(\tau) R(\tau) \log \pi(\tau)$$

$$= R_\mu^+ \sum_{\tau \in T^+} \mu_R^+(\tau) \log \pi(\tau) - R_\mu^- \sum_{\tau \in T^-} \mu_R^-(\tau) \log \pi(\tau)$$

$$= -R_\mu^+ \mathrm{KL}(\mu_R^+ \,\|\, \pi) + R_\mu^- \mathrm{KL}(\mu_R^- \,\|\, \pi) + C_1.$$

Putting it all together, we see that

$$\mathcal{L}_{\mu,c}(\pi) = C + R_\mu^+ \mathrm{KL}(\mu_R^+ \,\|\, \pi) - R_\mu^- \mathrm{KL}(\mu_R^- \,\|\, \pi) - c \mathrm{KL}(\mu \,\|\, \pi),$$

with $C$ a constant independent of $\pi$.

### G.2   Proof of Proposition 3.2

We start by showing that $\rho(x, 0, b) \geq 0$ for all $x \geq 0$ and all $b > 0$. If $x < b$, then $\rho(x, 0, b) = x > 0$. If $x > b$, then $\rho(x, 0, b) = b \left(1 + \log \frac{x}{b}\right) \geq 0$ since $\frac{x}{b} \geq 1$.

Then, starting from

$$J_{\mathrm{TOPR}}(\pi) = \sum_{\tau \in T^+} \mu(\tau) \rho \left( \frac{\pi(\tau)}{\mu(\tau)}, a^+, b^+ \right) R(\tau) + \sum_{\tau \in T^-} \mu(\tau) \rho \left( \frac{\pi(\tau)}{\mu(\tau)}, 0, b^- \right) R(\tau), \qquad (8)$$

we see that the second term is negative as the sum is over trajectories with negative rewards and both $\mu(\tau)$ and $\rho \left( \frac{\pi(\tau)}{\mu(\tau)}, 0, b^- \right)$ are positive.

Hence,

$$J_{\mathrm{TOPR}}(\pi) \leq \sum_{\tau \in T^+} \mu(\tau) \rho \left( \frac{\pi(\tau)}{\mu(\tau)}, a^+, b^+ \right) R(\tau)$$

$$\leq \sum_{\tau \in T^+} \mu(\tau) \frac{\pi(\tau)}{\mu(\tau)} R(\tau) \qquad \text{(since } \rho\left(x, a^+, b^+\right) \leq x\text{)}$$

$$\leq \max_\tau R(\tau) \qquad \text{(since } \pi(\tau) \leq 1 \text{ for discrete trajectories)}$$

$$= B \,.$$

This concludes the proof.

---

[5]If $R_\mu^+ = 0$ (resp. $R_\mu^- = 0$), our argument holds for any $\mu_R^+$ (resp. $\mu_R^-$).

