# OpenReview forum: "Tapered Off-Policy REINFORCE - Stable and efficient reinforcement learning for large language models"
_NeurIPS.cc/2025/Conference — NeurIPS 2025 poster_

### Official Review · Reviewer_3Rpm · 2025-07-02

**Clarity:** 4
**Significance:** 3
**Originality:** 3
**Rating:** 5
**Confidence:** 3

**Summary:**

This work introduces a new RL algo called Tapered Off-Policy REINFORCE (TOPR) to address the instability issues of REINFORCE even when applied in off-policy settings (for fine-tuning large language models). TOPR uses asymmetric importance sampling with different truncation parameters for positive and negative examples, enabling fast and stable learning (without requiring KL regularization).

**Questions:**

none

**Ethical Concerns:**

["NO or VERY MINOR ethics concerns only"]

**Final Justification:**

no revisions needed.

**Limitations:**

ran only on small-ish networks

**Paper Formatting Concerns:**

- In figure 5, the legend (and how things are ordered) makes it hard to read.

**Quality:**

4

**Strengths And Weaknesses:**

Strengths:
- Rigorous theory
  - algo has intuitive-ish justification
  - analysis through prop 3.1 and 3.2, showing that TOPR optimizes a bounded objective when a = 0 is insightful
  - asymmetric treatment of positive vs negative examples is well-motivated
  - simple compared to other RL algos
- seems to work really well compared to others
   - stable off-policy learning without requiring KL regularization
   - effectively uses both positive and negative examples -
   - simple to implement - just a modification to standard REINFORCE

Weaknesses
- I would have liked to see more comparisons to more RL algos. Other baselines could be: TRPO, PPO, GRPO.
- wasn't run on large networks

---

> ### Author Rebuttal · Authors · 2025-07-29
>
> Thank you for the kind review. Regarding your two comments:
>
> 1/ Comparison to other methods
>
> - We *do* compare to PPO: theoretically in Figure 2 and empirically in Figure 1.
> - While GRPO uses advantages, it does not address the main limitation of PPO which is that the gradient does not flow as soon as the policy deviates significantly from the base policy, making it unsuitable for truly off-policy optimization. We can thus expect to observe the same suboptimal plateau as PPO.
> - Although TRPO can be faster than PPO in terms of updates, making sure the constraint is satisfied can be computationally expensive.
>
> 2/ Larger networks
>
> We appreciate the desire for more thorough experiments, a desire echoed by other reviewers. Sadly, the current set of experiments was already done at significant expense, especially given their thoroughness, for instance to evaluate the influence of the baseline. Larger networks also achieve high accuracy on MATH and GSM8K, leading to most examples being positive. Hence, a meaningful experiment using larger networks should involve harder datasets, leading to even larger compute requirements.
>
> However, given the strong demand for the reviewers for such additional tasks, we would love if you collaboratively could have a prioritized list of tasks that should be added. This would allow us to strengthen the submission while keeping computational costs manageable.
>
> 3/ Figure 5
>
> If you have the time, we would love to know how we can improve Figure 5. Are you talking about the right plot, where the legend crosses the lines? In any case, we will update the Figure to improve its readability.

---

> > ### Comment · Reviewer_3Rpm · 2025-08-01
> >
> > Thank you for your thoughtful response
> >
> > re 3/ Figure 5 Just updating the legend to be ordered the same would help a lot.
> >
> > re 2/ Larger networks qwen has a good 14B and 32B network to work with

---

> > > ### Author Response · Authors · 2025-08-06
> > >
> > > Thank you for the precision for Fig. 5.
> > > We will take your suggestion regarding qwen into consideration but, once again, we cannot promise we will have the budget for it.

---

### Official Review · Reviewer_5Gwa · 2025-07-03

**Clarity:** 4
**Significance:** 3
**Originality:** 4
**Rating:** 5
**Confidence:** 3

**Summary:**

The authors introduce Tapered Off-Policy REINFORCE (TOPR), a new algorithm for fine-tuning large language models with reinforcement learning that remains stable even when operating off-policy and does not require KL regularization. TOPR consists of an asymmetric importance-sampling update: positive trajectories receive a supervised fine-tuning–style weight independent of the current policy, while negative trajectories are handled with a truncated importance-sampling correction to prevent destructive gradient updates.Theoretical analysis shows that this “tapered” scheme bounds the learning objective and avoids the need for baselines or KL penalties that can mask valuable negative-feedback information

Empirically, the authors demonstrate on GSM8K and MATH reasoning benchmarks that TOPR outperforms REINFORCE, PPO, and DPO. Also, in a multi-iteration fine-tuning setup, TOPR combined with “Anna Karenina” sampling (which prioritizes diverse negative examples and keeps just the first correct solution) enables a DeepSeek 8 B-parameter model to exceed the performance of much larger Llama 70 B-parameter model over successive rounds of training.

**Questions:**

* The paper evaluates TOPR only on GSM8K and MATH reasoning tasks. Could you please include additional tasks to demonstrate generality? Could be code generation, or dialog based tasks or even non-LLM traditional RL tasks.
* Could you please report the run-time comparison for TOPR vs PPO, DPO, REINFORCE?

**Ethical Concerns:**

["NO or VERY MINOR ethics concerns only"]

**Final Justification:**

I'm happy with the rebuttal and maintain my positive score.

**Limitations:**

Yes

**Quality:**

3

**Strengths And Weaknesses:**

### Strengths:
* The paper introduces a “tapered” importance-sampling update that treats positive examples like supervised fine-tuning while truncating negative weights. This is a novel unification of SFT and off-policy RL .
* The algorithm is presented cleanly and its behavior intuitively illustrated, making both motivation and mechanics easy to follow.
* The paper offers good theoretical analysis, proving that the TOPR objective is bounded under its tapered importance-sampling scheme, ensuring stable off-policy updates without requiring KL penalties.
* Across GSM8K and MATH benchmarks, TOPR consistently outperforms strong baselines (REINFORCE, PPO, DPO).

### Weaknesses:
* Experiments are limited to two mathematical reasoning tasks (GSM8K, MATH), leaving the method’s applicability to broader language tasks untested.

---

> ### Author Rebuttal · Authors · 2025-07-29
>
> Thank you for the review.
>
> 1/ More tasks
>
> We acknowledge your desire, echoed by other reviewers, to see TOPR tested on more standard benchmarks. Unfortunately, our GPU budget (in dollars) required us to make the choice between shallow evaluation across many datasets, or focusing on a smaller set and going in depth. The current set of experiments was already done at significant expense given their thoroughness, and every additional task requires careful tuning of the baselines to make sure our results are trustworthy.
>
> However, while our submission focused on math reasoning datasets, we also did preliminary evaluations of TOPR on query expansion. In these experiments, the website clinicaltrials.gov was used as a ground truth for expanded query terms. For instance, a researcher may search "IV" but expect "intravenous" to be included in their search results. We fine tuned a Llama 3 8b model to produce a domain-specific set of query expansions after having collected and verified a corpus of positive and negative samples.
>
> TOPR was able to shift the increase the performance of the LLM, measured as the intersection over union of the expanded set of queries generated by the LLM compared to the ground truth. This was done without any domain knowledge. The results remain preliminary but could be included if desired, although they do not correspond to a standard benchmark.
>
> 2/ Compute budget
>
> All methods that we evaluated require the same compute budget for generating samples.
>
> For training, TOPR, PPO and REINFORCE all require about the same time to perform one gradient step (on a H100 node in our setup, about 1.3s per batch). PPO and TOPR compute the ratio of probabilities from the reference model, which requires an additional forward pass (the reference). We found this to add a small overhead (5-10% of per-step cost), with computing the gradients and the backward pass the dominant cost. DPO is slower because it performs the backward pass on two samples and computes two reference probabilities.
>
> Overall, every figure that uses "Gradient steps" as the x-axis can be equivalently thought of as approximately using runtime on that axis.

---

> > ### Comment · Reviewer_5Gwa · 2025-08-04
> >
> > Thank you for your response. I would suggest adding the query expansion results in the paper as well, even if they do not correspond to a "standard" benchmark. I think it'll give the readers a sense of how generalizable TOPR is beyond just mathematical reasoning tasks.

---

> > > ### Author Response · Authors · 2025-08-06
> > >
> > > Thank you for the suggestion, the extra page should allow us to comfortably add that experiment.

---

### Official Review · Reviewer_cD7b · 2025-07-03

**Clarity:** 3
**Significance:** 3
**Originality:** 3
**Rating:** 4
**Confidence:** 2

**Summary:**

This paper introduces Tapered Off-Policy REINFORCE (TOPR), an algorithm for RL fine-tuning of LLMs that addresses the issues of other algorithms in the off-policy setting, such as instability and slow convergence. The idea of the method is to update on positive-reward examples without importance sampling (enabling fast learning even when unlikely under the current policy), and use truncated importance sampling (TIS) on negative-reward examples (preventing destructive parameter updates as the policy diverges from the data distribution). The authors demonstrate that TOPR outperforms naive REINFORCE, PPO, and DPO in the off-policy setting on GSM8K and MATH reasoning benchmarks; they also show that TOPR is more robust to dataset imbalance compared to doing TIS on all examples. Other contributions include theoretical analysis of TOPR (boundedness of the objective and stability), a reinterpretation of the REINFORCE baseline parameter through the lens of effective dataset composition, and the effectiveness of a new data sampling strategy which focuses on negative examples.

**Questions:**

- In a head-to-head comparison with prior art (TIS) on Figure 4, TOPR is shown to be much more robust when effective % of positive examples is small. Would this hold up if we compare TOPR with TIS+downsampling, where we downsample the negative examples to increase % of positives and only then do TIS on it? It is true that some negative examples would be thrown out doing this, but it would be great to research whether this waste would in fact push the performance of TIS significantly below the performance of TOPR in this setting, and by how much.
- Would Figure 4 look the same with x-axis being the actual proportion of positive examples in the dataset?
- Table 1 claims that TOPR is faster than TIS - is there experimental evidence to support this?
- What number of gradient accumulation steps are you using and have you tried different ones? My concern is that it's plausible it could significantly impact the robustness of different methods to the dominance of negative examples in the dataset.
- The paper says "All main claims of the paper are supported by experiments containing error bars." and describes the bootstrap procedure, but I can't see the error bars in any figures—is it because the confidence intervals are too small to be rendered?

**Ethical Concerns:**

["NO or VERY MINOR ethics concerns only"]

**Final Justification:**

Clearly reasons to accept outweigh reasons to reject, but with a margin that is not very high:
- It's a new principled and elegant method that seems to work better than others on an important class of problems;
- The evaluation, however, didn't allow me to fully confirm the superiority of the new method. Many concerns were addressed by the authors, and I find myself directionally agreeing with their arguments, but they still didn't fully allow me to validate the evaluation methodology and results. If it wasn't for this, I would have put 5/6, because the other parts of the work are solid.

**Limitations:**

yes

**Paper Formatting Concerns:**

No issues

**Quality:**

3

**Strengths And Weaknesses:**

Strengths:
- Justifications of TOPR and comparisons with other similar methods in Sections 2 and 3 are clear and insightful, and help to understand and appreciate the new method. They are corroborated by extensive theoretical analysis.
- TOPR has a number of advantages—it's designed in a very principled way, it's simple to implement, doesn't require regularization, and is shown to be more robust than alternatives off-policy.

Weaknesses:
- The experimental validation is thorough but I feel like it has not fully convinced me of superiority of TOPR for the following reason. The closest method to TOPR is TIS, as TOPR only differs from it by removing the density ratio for positive examples. It also seems that TIS is also the closest to TOPR in performance, performing better than DPO and PPO. However, the only head-to-head comparison of TIS and TOPR is on Figure 4; it shows performance of these 2 methods for different values of the REINFORCE baseline. It strongly hints at greater robustness of TOPR when negative examples dominate the data. However, I didn't understand why this setting of low effective (not actual) percentage of positives would be a particularly important one in practice. It is also not fully clear to me if this performance of TIS in the setting of 1-2% positives is the best we can do with TIS. Finally, the core value proposition of TOPR seems to be that it avoids "waste" of ignoring or throwing out negative examples, but has the pure cost of this waste been computed experimentally, in isolation from differences in the behavior of different methods?

---

> ### Author Rebuttal · Authors · 2025-07-29
>
> Thank you for your review and many questions.
>
>
> 1/ Comparison between TOPR and TIS for various proportions of positive examples
>
> While we did not perform the experiment for TIS, Figure 8 in the appendix gives us insight on the relative impact of downsampling negative examples compared to downweighing them for TOPR. We can see that using 10% positive examples in the dataset performs comparably, albeit slightly better, to using 50% positive examples, regardless of the *effective* percentage of positive examples. Difference is on the order of a couple percentage points, which is less than the difference between TIS and TOPR shown in Figure 4. Should you place this as high priority (because of the incurred cost), we could run the equivalent of Figure 8 for TIS in the final version of the paper.
>
> Figure 8 should also give an idea of what Figure 4 would look like with the actual proportion of positive examples on the x-axis.
>
> Another big difference between TIS and TOPR comes when the dataset is not composed of trajectories generated from the base model but either from a more powerful model (distillation) or with expert trajectories. In the former case, the importance ratio can be close to 0 if the teacher model is very different from the student model and TIS will assign a very small gradient to these positive examples. In the latter case, one does not even have access to the sampling distribution and cannot use the importance ratio. We believe this is why, even though TIS performs well in our setup, SFT is the predominant method for fine-tuning LLMs.
>
> As your comment was echoed by all other reviewers, we will strongly emphasize this difference in the final version should our submission be accepted.
>
> 2/ Claims of Table 1
>
> Before discussing evidence, we want to highlight the fact that there would only be a significant difference between TOPR and TIS in the presence of desirable trajectories with low probability under $\pi$. We identify two cases where this can happen:
> - First, when most of the datasets is comprised of negative examples, the optimization will decrease $\pi(\tau)$ for sampled trajectories $\tau$, which are generally those with larger $\mu(\tau)$ with $\mu$ the sampling policy. Hence, during the stochastic optimization, we might select positive trajectories $\tau$ for which $\pi(\tau)$ is much smaller than $\mu(\tau)$. The gradient for these trajectories will be much smaller for TIS than for TOPR, leading to poorer optimization. The evidence for this is in Figure 4, where the performance of TIS drops much more than that of TOPR for low effective percentage of positive examples.
> - Second, when the dataset is composed of examples not drawn from the base model, for instance in the case of behaviour cloning. In that instance, even when there are only positive examples in the dataset, it might happen that $\pi(\tau)$ is much smaller than $\mu(\tau)$ and the model will not learn them. The empirical evidence for the superiority of TOPR over TIS in that context is that behaviour cloning uses an SFT-like objective, also used by TOPR for positive examples, and not the REINFORCE based one used by TIS.
>
> However, in cases where a/ the dataset is generated from the base model and b/ there is a signification proportion of positive examples in the dataset, we can expect $\pi(\tau)$ to never be much smaller than $\mu(\tau)$ and for TIS and TOPR to perform similarly, as observed on the right-hand side of the two plots in Figure 4.
>
> 3/ Gradient accumulation
>
> That is an interesting question. As experiments ran on 8 GPUs, we accumulated 1 gradient per GPU before applying the update (see section C.2 in the appendix). We did not try other values of gradient accumulation for this submission. However, we are in the process of open-sourcing a version of the code and we tested it with various batch sizes without observing any significant difference so far. Please note however that our experiments with that new codebase are not as thorough as the ones we have done for the submission.
>
> Could you maybe explain why you think different methods might be affected differently by the percentage of negative examples? We have additional theoretical results, which we decided not to include in the submission for conciseness, showing that TOPR is likely to have lower variance than SFT as it minimizes the impact of a gradient step on trajectories not in the batch. We can definitely consider including these results in the Appendix should you think this is important.
>
> 4/ Error bars
>
> All figures except Figure 6 and Figure 9 (in the Appendix) contain error bars corresponding to two standard errors (see section C.2). Error bars are indeed extremely small due to the number of solutions generated. Note that our error bars capture uncertainty in the generation of the test examples, not in the training process nor the creation of the training set as this would significantly increase the total computational cost of the experiments.
>
> 5/ Cost of discarding negative examples
>
> The sole difference between SFT and TOPR is the inclusion of negative examples. Hence, the cost of this waste cannot be isolated from "differences in the behavior of different methods" as the different behavior between SFT and TOPR is *precisely* this waste. This is in contrast with, e.g., the difference between TOPR and Naive REINFORCE where all examples are included but with a different update. Thus, Figure 3 highlights this waste.

---

> > ### Comment · Reviewer_cD7b · 2025-08-05
> >
> > Thanks for a detailed reply and clarifications! This allowed me to understand the paper better and I agree with most of the points.
> >
> >
> > > While we did not perform the experiment for TIS, Figure 8 in the appendix gives us insight on the relative impact of downsampling negative examples compared to downweighing them for TOPR. We can see that using 10% positive examples in the dataset performs comparably, albeit slightly better, to using 50% positive examples, regardless of the effective percentage of positive examples.
> >
> > I see. So after you downsampled negative examples in the 10% positive examples dataset to get a 50% positive examples dataset, the performance drop was comparable to what you see with varying % of effective positive examples in that range. This allows you to say varying effective examples % has similar performance effects to varying actual examples %.
> >
> > > Difference is on the order of a couple percentage points, which is less than the difference between TIS and TOPR shown in Figure 4.
> >
> > This is correct, but that TOPR-TIS difference is seen at the extreme end of the scale. If I understand correctly, it's plausible that the correspondence between varying effective vs actual % could break down at that end of the scale where it wasn't established. This would be problematic for transfer of the results with effective % positives variation to actual % positives variation.
> >
> > By the way, there is one inconsistency I see on this page which I can't explain, maybe I'm missing something. Figure 8 left pane shows downsampling negative examples to the point of only leaving the positives (100% effective % of positives) lets pass@1 drop by 10+ percentage points. If varying effective positives % substitutes for varying actual positives %, then should we also see this low performance on the SFT plot (Figure 7 upper left)?
> >
> >
> > On "Claims of Table 1": you arguments here for why TOPR would be faster than TIS totally make sense. However, my point here is that if Table 1 claims it converges faster, it would be good to see numbers or plots confirming this, not only arguments.
> >
> >
> > > Another big difference between TIS and TOPR comes when the dataset is not composed of trajectories generated from the base model but either from a more powerful model (distillation) or with expert trajectories. In the former case, the importance ratio can be close to 0 if the teacher model is very different from the student model and TIS will assign a very small gradient to these positive examples.
> >
> > Agreed! This might be a very fruitful setting to explore experimentally.

---

> > > ### Author Response · Authors · 2025-08-07
> > >
> > > Thank you for your kind acknowledgement of our reply and the insightful followup questions.
> > >
> > > The difference between TOPR and TIS is for 1%, 2%, and 5% effective positive samples. While it is indeed possible that the correlation between actual and effective percentage breaks down at that end of the scale, we do not consider this the most plausible explanation for two reasons:
> > > - The correlation between TOPR-10p and TOPR-50p is still very high at that end (Fig. 8, left). This would mean that the correlation would break only for TIS.
> > > - Going for 10% actual to 5% effective is not a strong shift so the correlation would have to fall dramatically faster on the left end of the plot than the right end.
> > >
> > > You are right that the two numbers (end performance for SFT vs 100% effective positives) are different. One explanation is the diversity of examples in the training datasets, even at equal numbers of training updates. For example, the 50p dataset in Figure 8 “performs” worse than the 10p dataset with various choices of baseline, but the overall shape remains (optimum at <= 20% effective positives). Similarly, Fig. 3 uses a total of 119k data points (7473 x 16), whereas Fig. 4 uses 50k sampled from those, to keep computational costs manageable. That 50k is not chosen at random from the 119k since we fix the proportion of positives to 10% (versus ~50% for the full dataset).
> > >
> > > The other consideration is the effect on the optimization process as a whole, since our theoretical analysis only considers the expected gradient. Changing the baseline changes the weight of positive and negative examples, with c = -1.0 resulting in a weight of 0 for negative examples and 2 for positive examples. So a minibatch with eight examples and just one positive example will have a total weight of 2, rather than 8 when c = 0. Although this is “working as intended”, one can imagine that gradient clipping, second order optimization, etc. also plays a role. Note that when p = 10%, we achieve \tilde p = 50%, 80% with c = -0.8, c = -0.945, so the effect exists even for smaller values of \tilde p. However, we believe these effects to be smaller than the one induced by the smaller dataset given the similar performance of TOPR-10P and TOPR-50P.
> > >
> > > We will clarify that the relationship between baseline and effective positives is approximate at the extremes, and obviously is also affected by the underlying sampling regime.

---

### Official Review · Reviewer_PUSX · 2025-07-06

**Clarity:** 2
**Significance:** 2
**Originality:** 2
**Rating:** 4
**Confidence:** 3

**Summary:**

This paper proposes Tapered Off-Policy REINFORCE (TOPR), a policy gradient algorithm designed for the stable and efficient off-policy fine-tuning of large language models. The method uses an asymmetric update rule, applying a SFT objective to positive-reward examples and a truncated importance sampling objective to negative-reward examples. The authors claim this approach ensures stable learning dynamics, even in highly off-policy scenarios, without requiring an explicit KL regularization term.

**Questions:**

1.  Could you please provide precise details on how the dataset (with absolute rewards) was converted into preference pairs $(yw, yl)$ for the DPO baseline? How were the "up to 16 contrastive pairs" per prompt constructed? A fair comparison is essential to validate TOPR's superiority.
2. Your analysis of the baseline $c$ is insightful. However, how should a practitioner select $c$ to achieve the optimal 10-20% effective positive ratio for a new dataset or model without running an expensive hyperparameter sweep?
3.  The canonical TOPR algorithm uses an SFT-style update for positive examples ($a+ = 1$). What happens if you use a truncated importance sampling update for positive examples as well (e.g., $a+ = 0, b+ = C$ for some $C>1$)? Does this harm performance, or could it provide a better bias-variance trade-off?
4.  Have you explored TOPR's performance on tasks with denser or non-binary reward signals, such as summarization (e.g., optimizing for ROUGE scores) or dialogue systems? How does it perform when most examples have a reward between -1 and 1?

**Ethical Concerns:**

["NO or VERY MINOR ethics concerns only"]

**Final Justification:**

Concerns have been addressed.

**Limitations:**

Yes

**Quality:**

2

**Strengths And Weaknesses:**

Strengths
1. The core idea of TOPR—asymmetrically handling positive and negative examples—is elegant, simple to implement, and well-motivated. It cleverly combines the speed of supervised learning with the stability of truncated importance sampling.
2. The paper offers a compelling re-interpretation of the REINFORCE baseline's role in the off-policy setting. Framing it as a mechanism to control the effective ratio of positive-to-negative examples, rather than purely for variance reduction, is a potentially valuable contribution to the community's understanding.

Weaknesses
1. The TOPR algorithm is presented as a new contribution. However, a deconstruction of its gradient update reveals that it is a straightforward composition of two well-established techniques. The update for positive examples is identical to reward-weighted supervised fine-tuning (SFT), a method used in prior work like STaR[1]. The update for negative examples is a direct application of truncated importance sampling [2], a known technique for controlling variance in off-policy estimation.
2. The paper convincingly argues that the REINFORCE baseline $c$ is critical for controlling the "effective proportion of positive examples". However, it provides no practical guidance on how a user should set this crucial hyperparameter, short of an expensive experimental sweep, which undermines the algorithm's claimed simplicity.

[1] STaR: Self-Taught Reasoner Bootstrapping Reasoning With Reasoning

[2] Importance Resampling for Off-policy Prediction

---

> ### Author Rebuttal · Authors · 2025-07-30
>
> We thank the reviewer for their comments and questions.
>
> While we agree that TOPR can be viewed as a mixture of SFT (for the positive examples) and TIS (for the negative examples) on self-generated examples, we are a bit surprised to see this listed as a weakness rather than a strength. Had such a mix been straightforward, one would expect this method to have been developed earlier given its strong results. Most implementations that only rely on positive rewards, such as STaR, ReST, etc. emphasize complex prompting techniques (such as rationalization) and dataset curation, while here we deal with the negative samples directly.
>
> Additionally, the derivation of TOPR is not a random idea. It is the result of an understanding of the behavior of several well-established methods and their own strengths and weaknesses. We are glad to have made our presentation clear enough that the idea of TOPR seems obvious but we dispute the claim that it is not a new contribution.
>
> On the other hand, we agree with the statement that we provide no way to easily set the baseline parameter. As we stated in our submission, we found the best effective percentage to be around 10-20% for both GSM8K and MATH. Of note, our result shows that the GRPO baseline is not necessarily optimal. A deeper understanding of the baseline parameter would indeed be most welcome.
>
> Moving on to your questions:
>
> 1/ Construction of the dataset for DPO
>
> For each question, we generate 16 answers, leading to two sets $T_+$ and $T_-$. We then sample with replacement 16 contrastive pairs for each question. When one of the two sets is empty, i.e. all answers are incorrect or all answers are correct, then we cannot generate contrastive pairs and the question is discarded. We do this to compare things fairly across algorithms (same number of generated responses). Although we could attempt a different scheme (e.g., sample 8 distinct pairs for each question), this would be expensive for questions for which the model rarely generates the correct answer.
>
> 2/ Setting of the baseline
>
> There is an explicit formula for the setting of the baseline parameter to achieve a particular effective positive ratio, shown in Eq. 7 in Appendix B. If the proportion of positive examples in the dataset is $p$ and the baseline is $c$, then the effective proportion of positive examples is $\tilde{p} =\frac{p(1-c)}{1 + c(1 - 2p)}$. Solving for $c$ readily gives the baseline parameter to use. Not that this is for rewards of +1 and -1 but the formula can be adjusted for any rewards.
>
> 3/ TIS vs. TOPR
>
> The importance of the SFT update for the positive examples, as opposed to simply using TIS for all examples, is of prime importance. This concern was echoed by other reviewers and we thus intend to greatly clarify the difference in the revised submission.
>
> Before answering your question, we want to clarify what might have been a misunderstanding: the method we call TIS in our submission uses truncated importance sampling for both positive and negative examples, as you suggest.
>
> We were also surprised by the strong performance of TIS given the predominance of SFT-like methods to finetune LLMs. This said, TIS does struggle when there are few positive examples, because in this case their importance ratio goes to zero faster than the model can positively reinforce those examples.
>
> TIS requires a reference or sampling distribution for the positive examples, allowing it to leverage expert data for instance. As such expert trajectories are often used to do behaviour cloning, TIS is hardly used despite its strong performance when the dataset is entirely generated from the base model. Further, even if we knew the sampling distribution for expert trajectories, some or many importance ratios for these trajectories would be close to 0 if the base model is far from an expert, leading to a small gradient for these trajectories. We would then likely be in a similar situation as that when there are few positive examples, i.e. TOPR would outperform TIS.
>
> In addition to this property, we did not find a setting where TIS consistently outperformed TOPR. As shown in Figure 4, there are cases where TIS is superior to TOPR, but they require the baseline to be very carefully selected, and thus a potentially expensive hyperparameter sweep.
>
> 4/ Non-binary rewards
>
> We decided to privilege a thorough experimental analysis on a smaller set of tasks rather than provide less conclusive answers on more tasks. We acknowledge the compromises of this decision and we do not have results for non-binary reward signals. Should the reviewer have suggestion on which such dataset, with manageable computational complexity, to include, we will give it strong consideration.

---

> > ### Comment · Reviewer_PUSX · 2025-08-07
> >
> > Thanks for answering my questions. My concerns have been addressed.

---

### Decision · Program_Chairs · 2025-09-17

**Decision:**

Accept (poster)

**Comment:**

This paper presents a relatively simple idea and this is also its biggest weakness.
It adapts the REINFORCE
algorithm to use importance sampling when the reward is negative and
the unaltered algorithm when it is positive.  This gives it some nice
properties including the ability to use positive and negative examples
and the ability to operate stably off-policy.  The paper is well
written and achieves improved results on GSM8K and MATH benchmarks
with a Llama 8B model.  A particular strength of the paper is its
clear and insightful description of the problem and solution method.